# Vibration source analysis and structural optimization design of rotary table based on OTPA

**Miao Xie, He Wang◉\*, Zhixiang Liu, Suning Ma, Xia Wu, Yufeng Dong**

School of Mechanical Engineering, Liaoning Technical University, Fuxin, China

\* 179143021@qq.com

**Data Availability Statement:** All relevant data are within the manuscript.

**Funding:** This work was supported by the National Natural Science Foundation of China ( 51874158 ), the National Natural Science Foundation of China

## Abstract

Due to the influence of coal rock shape, hardness, working environment and other factors in the cutting process of cantilever roadheader, the cutting head will produce irregular and violent vibration. As the rotary table of key stress components, its operation process stability, dynamic reliability and life affect the cutting efficiency and cutting stability of cantilever roadheader. In order to study the vibration characteristics of the rotary table in the cutting process, firstly, based on the theory of spatial force analysis and calculation, the spatial mechanical model of the rotary table of the cantilever roadheader is established. By solving the balance equation of the rotary table force system, the variation law of the load at the hinge ear of the rotary table with the cutting pitch angle and the horizontal angle is obtained. Secondly, based on the path transfer analysis method of working condition, the vibration data of cutting head, cutting cantilever, cutting lifting and rotary hydraulic cylinder under stable cutting condition are taken as input signals. By constructing the transfer path analysis model of rotary table working condition, the synthetic vibration of rotary table in cutting process is simulated, and the main vibration source of rotary table is determined. Then, the vibration contribution and contribution degree of each vibration excitation point to the hinge ear of rotary table are studied. By building a cutting test bench, the vibration response of rotary table in cutting process is tested to verify the correctness of the theoretical model. Thirdly, based on the frequency domain analysis method of random vibration fatigue life, combined with the S-N curve of the rotary table, the PSD curve at the maximum stress of the rotary table is obtained by modal excitation method, and the load data is imported into ANSYS nCode software to obtain the life cloud diagram and damage cloud diagram of the rotary table, and then the fatigue life of the rotary table under symmetrical cyclic load is solved. Finally, based on the response surface optimization analysis method, the maximum stress and maximum deformation of the rotary table are taken as the optimization objectives, and the aperture of each hinge ear of the rotary table is taken as the optimization variable. Based on Design Expert, a second-order regression model is established to realize the multi-objective optimization design of the key stress parts of the rotary table in the cutting process. The simulation results show that under the same cutting conditions, the maximum stress of the optimized rotary table is reduced by 15.82% year-on-year, and the maximum deformation is reduced by 24.70% year-on-year. The optimized rotary table structure can

Youth Science Foundation ( 51904142 ), the Liaoning Provincial Department of Education ( LJ2019ZL003 ), and the China Huaneng Group Headquarters Science and Technology Funding Project ( HNKJ20-H34 ). Sponsor : XIE Miao provides design ideas and paper direction guidance. Sponsor : LIU Zhixiang assisted in the data collec-tion and analysis of the paper.

**Competing interests:** The authors have declared that no competing interests exist.

better adapt to the cutting process, which is beneficial to improve the service life of the rotary table and enhance its operation stability. The research results are beneficial to enrich the relevant research theory in the field of rotary table vibration of cantilever roadheader, and are beneficial to improve the service life of the rotary table and the efficiency of tunneling and mining.

## Introduction

Cutting vibration is common in the working process of roadheader [1–4]. The rotary table is most prone to fracture failure under irregular impact load, which affects the working efficiency and cutting stability of roadheader [5–7]. Therefore, in order to ensure the safe and stable work of the rotary table and realize the stability and accuracy of the cutting operation, it is of great engineering significance to analyze the vibration of the rotary table in the cutting process, study its vibration distribution law, vibration source and vibration optimization, and then reduce the impact of the cutting vibration on the rotary table in the cutting process.

In order to study the vibration characteristics of rotary table in cutting process, LIU et al. [8] took the EBZ135 roadheader as the research object, used the virtual prototype to model the simplified rotary and cutting mechanism, carried out the kinematics simulation, and obtained the stress state of the rotary table. MA et al. [9] used the finite element method to study the load distribution of the rotary table under extreme conditions, and accurately obtained the stress distribution that can fully reflect the mechanical properties of the rotary table. Based on the analysis results, the strength test and structural optimization of the structure of the rotary table are carried out. WU et al. [10] analyzed the stress and strain distribution of the rotary table of the cantilever roadheader during operation. The simulation results show that the main load of the rotary table comes from its lifting hinge ear and the ear connected to the cutting arm. After analyzing the stress and deformation generated under different thickness of the ear, it is found that increasing the thickness of the hinge ear can effectively reduce the stress and improve the safety of the rotary table.

In the study of the vibration control of the rotary table, the above scholars have carried out the vibration analysis of the rotary table in theory and software simulation. The above research has not carried out research on the vibration source of the rotary table, the distribution law of the vibration source and the vibration optimization of the rotary table, especially the vibration analysis of each hinge ear of the rotary table in the cutting process and how to reduce the vibration of the rotary table in the cutting process.

Operational transfer path analysis (OTPA) is an effective method to analyze the vibration source of mechanical moving parts. Under the running state of mechanical parts, the motion data are measured, and the transmissibility matrix of mechanical system from ' response-response ' is established to analyze the vibration source [11–15]. However, due to the signal interference between the input system data, the accuracy of data analysis will be affected [16]. As a matrix analysis method to effectively eliminate signal crosstalk and interference, singular value decomposition (SVD) is widely used in project engineering [17–19].

Random vibration fatigue is defined as the fatigue damage of the structure under random impact load [20]. The rotary table is in a random vibration environment for a long time during use, and the local structure will have a serious resonance response. The fatigue damage caused by this is one of the main forms of rotary table damage. Therefore, it is of great significance to analyze the fatigue life of the turntable under random dynamic load excitation.

At present, there are two most widely used methods for calculating the fatigue life of mechanical structures. One is the time domain analysis method, which is based on the rain flow counting theory. The other is the frequency domain analysis method, which is based on the power spectral density theory. The two methods have their own advantages. The time domain method is more accurate in calculating fatigue life. However, due to its cyclic method, the calculation amount is large, and the actual use is greatly limited [21–23]. The number of stress cycles in the frequency domain method is based on the approximate estimation of the stress power spectral density and the distribution density function. It does not require cycle counting, has a small computational workload, is easy to use, and has been more applied [24,25].

In summary, in order to reduce the vibration of the rotary table during the cutting process, the author considers the phenomenon that the rotary table on the cutting site is prone to failure and fracture. Based on the working condition path transfer analysis method, the vibration source of the rotary table during the cutting process is analyzed. The contribution and contribution degree of each excitation point of the cutting system to the vibration of the rotary table are studied, and the main excitation source causing the vibration of the rotary table is explored. A vibration test experimental platform is built to test the vibration response of each part of the rotary table during the cutting process. The finite element simulation calculation of the rotary table is carried out by using the frequency domain analysis method of random vibration fatigue life, and the life and damage cloud diagram are obtained. Finally, based on the response surface optimization analysis method, the multi-objective optimization design of the key force points of the rotary table is carried out to reduce the force and structural deformation of the rotary table during the cutting process, and the optimization of the rotary table structure is realized. The research results are beneficial to enrich the research theory of the vibration of the rotary table, and to improve the service life of the rotary table and the efficiency of excavation and mining.

## Mechanical properties analysis of rotary table

Taking the cutting head, cantilever, lifting, rotary cylinder and rotary table as the research objects, the mechanical model of rotary table in the cutting process of cantilever roadheader is established, as shown in Fig 1.

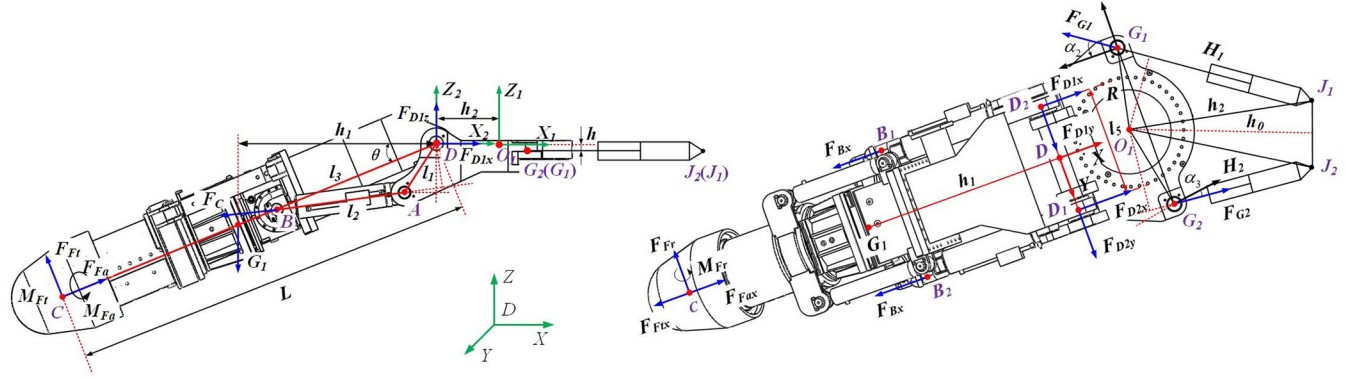

**Fig 1.**

According to the mechanical equilibrium equation $\Sigma F = 0, \Sigma M = 0$:

$$\begin{cases} F_{Fa} \cos\theta + F_{Ft} \sin\theta + F_{G2} \cos\alpha_3 - F_{G1} \cos\alpha_2 = 0 \\ -F_{Fr} + F_{Oy} + F_{G2} \sin\alpha_3 - F_{G1} \sin\alpha_2 = 0 \\ -F_{Fa} \sin\theta + F_{Ft} \cos\theta + F_{0z} - G_1 = 0 \end{cases} \quad (1)$$

$$\begin{cases} M_{Fa} \cos\theta + M_{Ft} \sin\theta + M_x + (F_{G1} \sin\alpha_2 - F_{G2} \sin\alpha_3)h = 0 \\ -M_{Fr} - F_{Fa} \cdot h_2 \sin\theta - F_{Ft}(L\cos\theta + H_2) + M_y + (F_{G1} \cos\alpha_2 - F_{G2} \cos\alpha_3)h + G_1(H_1 + H_2) = 0 \\ (F_{G1} \cos\alpha_2 + F_{G2} \cos\alpha_3)b_2 + (F_{G1} \sin\alpha_2 - F_{G2} \sin\alpha_3)b_1 - M_{Fa} \sin\theta + M_{Ft} \cos\theta - F_{Fr}(L\cos\theta + H_2) + M_z = 0 \end{cases} \quad (2)$$

Of which: $D_1$ and $D_2$ are the hinge points of cantilever and rotary table; A is the hinge point between the lifting cylinder and the rotary table; B is the hinge point of lifting cylinder and cantilever; C is the center of gravity of the cutting head; D is the center point of the hinge ear on the rotary table; $J_1$ and $J_2$ are the connection points between the rotary cylinder and the body frame; $G_1$ and $G_2$ are the center points of the left and right hinge ears of the rotary table; $F_{Ox}$, $F_{Oy}$, $F_{Oz}$ are the reaction force of the rotary bearing to the rotary table; $G_1$ is cantilever gravity; $M_x$ and $M_y$ are overturning moment; $M_z$ is friction torque; $F_{Dx}$, $F_{Dy}$ and $F_{Dz}$ are the three-way force of the hinge on the rotary table; Fc is the force of the lifting cylinder to the cantilever; $F_{G1}$ and $F_{G2}$ are thrust and tension of rotary cylinder; $H_1$ and $H_2$ are the length of rotary cylinder; $\alpha$ is the rotation angle; $\alpha_2$ and $\alpha_3$ are the angle between $F_{G1}$, $F_{G2}$ and X direction; $\theta$ is the pitch angle of the turntable; $M_{Fa}$, $M_{Ft}$, $M_{Fr}$ are cutting head torque; h is the vertical distance from the center of rotation to the center of rotation hinge; $h_1$ is the horizontal distance from the upper hinge ear to the center of gravity of the cantilever; $h_2$ is the horizontal distance from the center of rotation to D point.

Based on the established mechanical model, the relationship between the force of the hinge point of the cantilever and the rotary table with the pitch angle and the rotation angle is shown in Fig 2.

According to the results of the curve, the forces in the X, Y and Z directions of point D are not equal, the force in the X direction is the largest, and the force in the Y direction is the smallest. In the process of left and right yaw cutting, when the pitch angle $\theta$ is constant, the force of point D does not change much with the rotation angle $\alpha$, and $F_{D1x}$ is two ~ three times of $F_{D2x}$. When the rotation angle $\alpha$ is constant, the variation of $F_{D1x}$ and $F_{D2x}$ is opposite with the change of pitch angle $\theta$, and the variation is about 100 kN. The trend of $F_{D1z}$ and $F_{D2z}$ is consistent, and the change is about 200 kN.

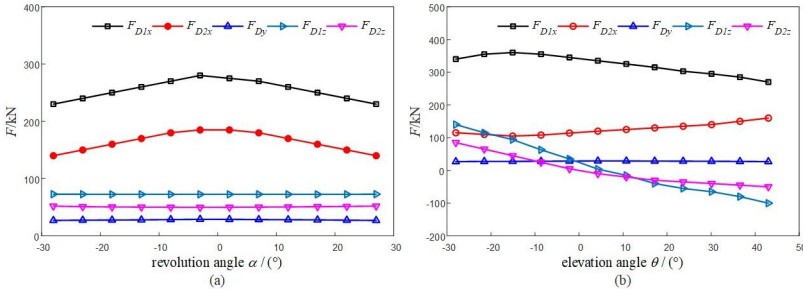

**Fig 2.**

## Working condition transfer path analysis model

Assuming that the system is a linear time-invariant system, the output vibration is linearly superimposed by the input vibration along each transmission path to the turntable [26,27]. The vibration transmission characteristics of the system are:

$$Y = XT \tag{3}$$

In the formula, X is the system input matrix, Y is the output matrix, and T is the transmissibility matrix.

In order to solve the transmissibility matrix T, m tests are required. Then X, Y and T in Eq (3) are:

$$Y = \begin{bmatrix} {}^{(1)}Y_1 & {}^{(2)}Y_2 & \cdots & {}^{(1)}Y_k \\ {}^{(2)}Y_1 & {}^{(2)}Y_2 & \cdots & {}^{(2)}Y_k \\ \vdots & \vdots & \ddots & \vdots \\ {}^{(m)}Y_1 & {}^{(m)}Y_2 & \cdots & {}^{(m)}Yk \end{bmatrix} \tag{4}$$

$$X = \begin{bmatrix} {}^{(1)}X_1 & {}^{(2)}X_2 & \cdots & {}^{(1)}X_n \\ {}^{(2)}Y_1 & {}^{(2)}Y_2 & \cdots & {}^{(2)}X_n \\ \vdots & \vdots & \ddots & \vdots \\ {}^{(m)}X_1 & {}^{(m)}X_2 & \cdots & {}^{(m)}X_n \end{bmatrix} \tag{5}$$

$$T = \begin{bmatrix} T_{11} & T_{12} & \cdots & T_{1k} \\ T_{21} & T_{22} & \cdots & T_{2k} \\ \vdots & \vdots & \ddots & \vdots \\ T_{n1} & T_{n1} & \cdots & T_{nk} \end{bmatrix} \tag{6}$$

In the formula, ${}^{m}Y_k$ represents the k th output of the system under the m th test condition, and ${}^{m}X_n$ represents the n th input of the system under the m th test condition.

When the number of test conditions is greater than or equal to the number of system inputs (k ≥ n), the input matrix X is invertible, and Eq (4) has a unique solution:

$$T = (X^T X)^{-1}(X^T Y) = G_{xx}^{-1} G_{xy} \tag{7}$$

In the formula, $G_{xx}$ is the self-power spectrum matrix of the system input, and $G_{xy}$ is the cross-power spectrum matrix of the system input and output.

When there is crosstalk between the transmission paths, the transmissibility matrix may have ill-posed problems. The input matrix is processed by SVD technology to eliminate the ill-condition of the input matrix and improve the recognition accuracy of the transmissibility matrix. The improved system input matrix is:

$$X^* = U\Lambda V^T \tag{8}$$

In the formula: U and V are unitary matrices of order m×m and n×n respectively; $\Lambda$ is a diagonal matrix of m × n, and the value $\sigma_i$ (i ≤ n) on the diagonal is the i th singular value of the input matrix X, satisfying $\sigma_1 \geq \sigma_2 \geq \ldots \geq \sigma_n \geq 0$.

Available from formulas (3) and (8):

$$T^* = V\Lambda^{*-1}U^T Y \tag{9}$$

Then the combined output signal $T^*$ can be expressed as:

$$Y^* = XT^* \tag{10}$$

In this contribution analysis, the contribution of the i th excitation point $X_i$ to the vibration of the target point $Y_j$ is:

$$\tilde{Y}_{ij} = X_i T^*_{ij} \tag{11}$$

The contribution is a vector signal with amplitude and phase, which is not convenient to compare the contribution of different excitation points. Therefore, the projection of the contribution expressed above in $\tilde{Y}_j$ direction is defined as the effective vibration contribution of the excitation point $X_i$ to the target point $Y_j$.

$$C_{ij} = |\tilde{Y}_{ij}| \cdot \cos\theta_{ij} \tag{12}$$

The effective vibration contribution represents the contribution of each excitation source to the vibration of the rotary table. Reducing the excitation source with a large vibration contribution is beneficial to reducing the vibration of the target point.

## Cutting test bench OTPA model

### Cutting test bench

The experimental research is carried out on the cutting test bench shown in Fig 3. The test bench consists of the experimental base platform, the body part, the rotary table, the cutting part, the cutting head, and the experimental coal wall. The experimental coal wall is configured according to the ratio of pulverized coal: cement: water = 1.62: 1: 0.49。 Because the test mainly depends on the acceleration sensor to realize the signal acquisition of mechanical action, this paper uses the YND-DR-3005 three-way acceleration sensor, the test pick selects

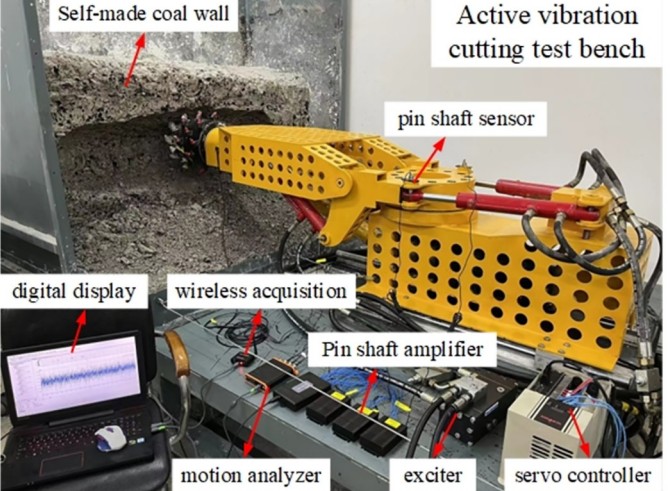

**Fig 3.**

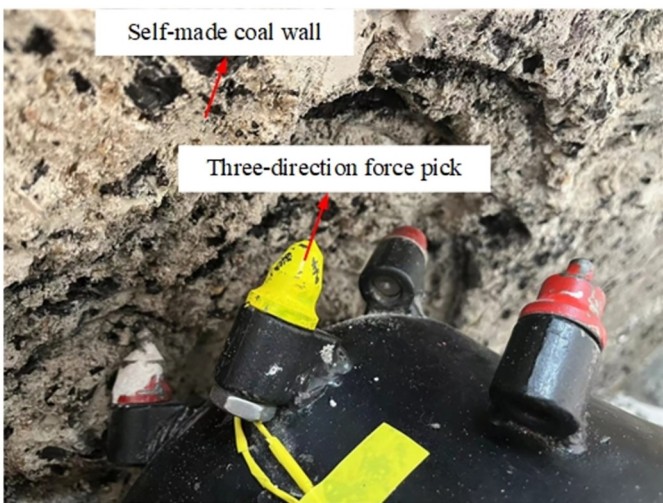

**Fig 4.**

the three-way force pick containing the strain flower, and Fig 4 is the installation position diagram of the three-way force pick during the cutting process; Fig 5 is the three-way acceleration sensor layout of the experimental process.

In this paper, a custom acceleration sensor and a HSGA1 series amplifier are used to measure the vibration of the rotary table during the cutting process. The experimental setting of the cutting head speed is 36 r / min, the cutting width is 150 mm, and the cutting depth is 50 mm. Customized acceleration sensor and HSGA1 series amplifier are used to measure the vibration of rotary table in cutting process. Data acquisition and analysis system includes Hua-soft-YL data acquisition system and DASP dynamic signal analysis system. In the simulation channel page of the test system software measurement mode, the sampling frequency is set to 8 KHz, and the number of sampling channels is set to 3. The vibration data of the turntable in X, Y and Z directions are measured respectively. The setting interface is shown in Fig 6.

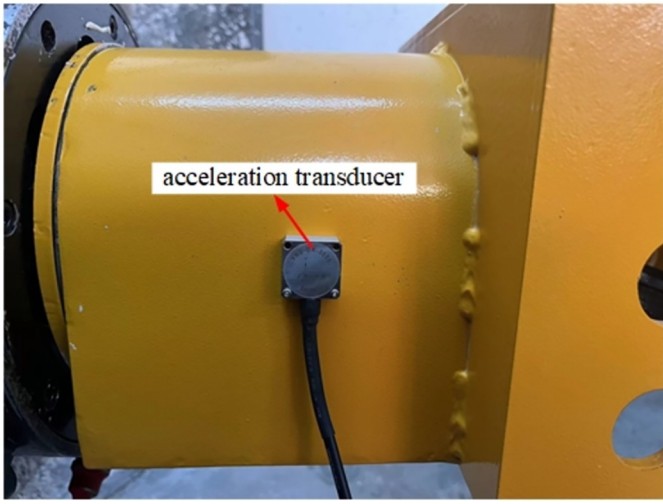

**Fig 5.**

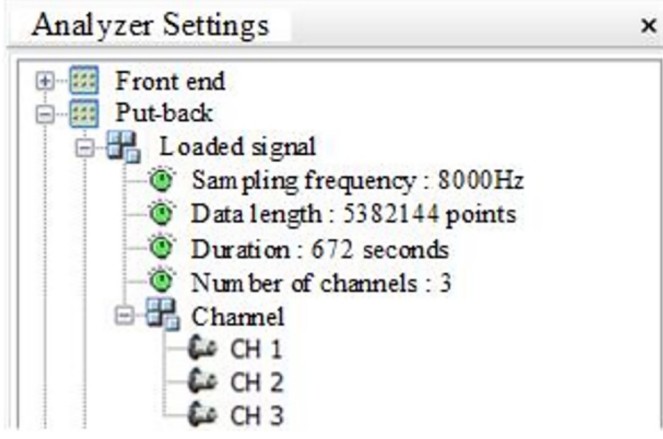

**Fig 6.**

The cutting test bench is started, and the test is cut from left to right, recorded every 20 seconds, a total of 10 cuts are performed, and a transverse coal wall section is cut. The data collector records the vibration of the rotary table in real time during the cutting process. The vibration of the rotary table in the vertical direction has the greatest impact on its structural damage, so the experimental test focuses on the vertical system of the rotary table.

## OTPA model construction

Under the customized test conditions (rated speed from top to bottom cutting), the vibration of the rotary table comes from the vibration of the cutting head, the vibration of the cutting arm, the vibration of the lifting and rotating hydraulic cylinder. The vibration acceleration sensor is installed at the hinge ears of the cantilever and rotary table as the test device, and the data are recorded and analyzed by the supporting data acquisition and analysis system. The vibration measuring points are arranged as shown in Fig 7. Among them, A1 and A2 are located on the left and right rotary hydraulic cylinders, A3 is located on the lifting hydraulic cylinder, A4 is located on the right end surface of the front part of the cutting arm, A5 is located on the maximum diameter end surface of the rear part of the cutting head, B1-B4 is located on the pin shaft of the upper, lower, left and right hinged ear holes of the rotary table.

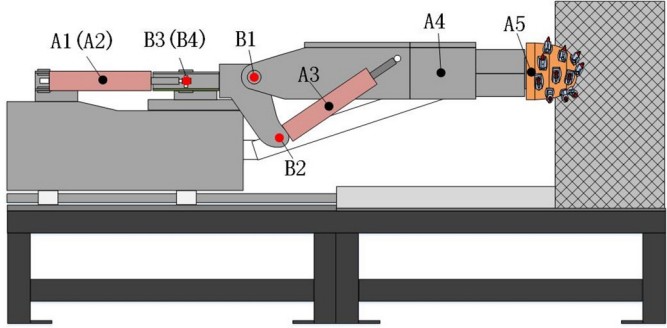

**Fig 7.**

A5 × 4 OTPA model is established with A1-A5 as the excitation point and B1-B4 as the target point.

## Model verification

In order to ensure that important excitation sources or transmission paths are not lost, coherence analysis is performed on the input data signals [28,29]. Fig 8 shows the coherence function of B1 measurement point under stable operation calculation condition. From the curve, the coherence coefficient of each target point of the turntable within 40 Hz is mainly between 0.85–0.95, and the measured signal is reliable, indicating that the important excitation source is not lost.

Based on the DASP dynamic signal acquisition and analysis system, the vibration characteristics of the rotary table under the above given working conditions are solved. The dynamic response analysis of the system is carried out at a time of 0 ~ 35 s. The vibration response of each hinge ear of the rotary table is obtained as shown in Figs 9–12.

The system transmissibility matrix is obtained by Eq (9), and the synthetic vibration response at the hinge ear of the rotary table is obtained by Eq (10). The measured vibration

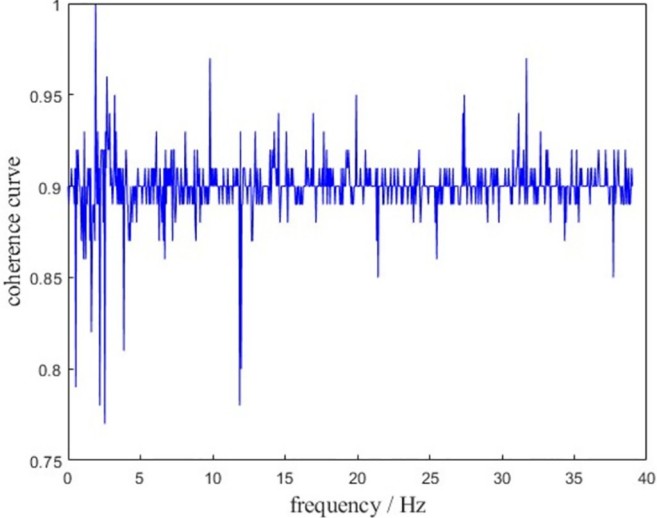

**Fig 8.**

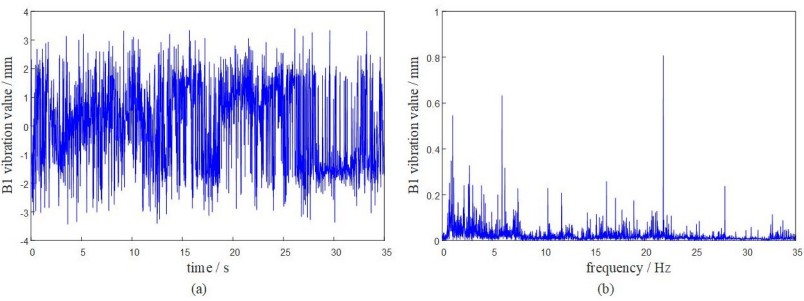

**Fig 9.**

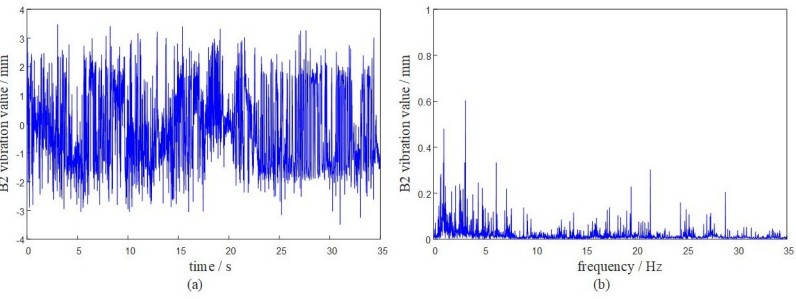

**Fig 10.**

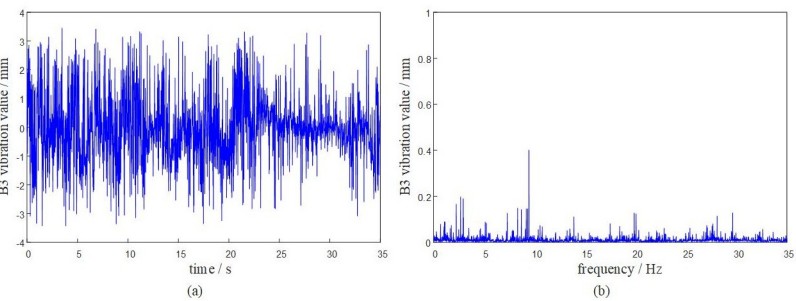

**Fig 11.**

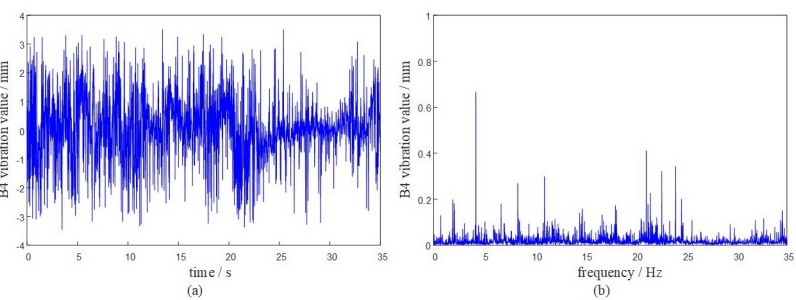

**Fig 12.**

data of the rotary table are shown in Table 1. Comparing the synthetic vibration and the measured vibration of the rotary table as shown in Fig 13, the error between the synthetic vibration and the measured vibration of each point is within 10%. The established OTPA model has high reliability and can accurately represent the vibration transmission characteristics of the rotary table.

It can be seen from the curve results that the maximum positive amplitude of the turntable is 3.5083 mm, and the maximum negative amplitude is -3.4915 mm. The turntable has a large peak near 20Hz-25Hz, indicating that the turntable has resonance phenomenon in this range.

**Table 1. Measured vibration data of the rotary table.**

| measuring point | B1 | B2 | B3 | B4 |
|---|---|---|---|---|
| maximum value /(mm) | 3.3950 | 3.4751 | 3.4512 | 3.5083 |
| minimum value /(mm) | -3.4321 | -3.4915 | -3.4287 | -3.4589 |
| mean value /(mm) | 1.2758 | 1.0971 | 0.8291 | 0.8317 |

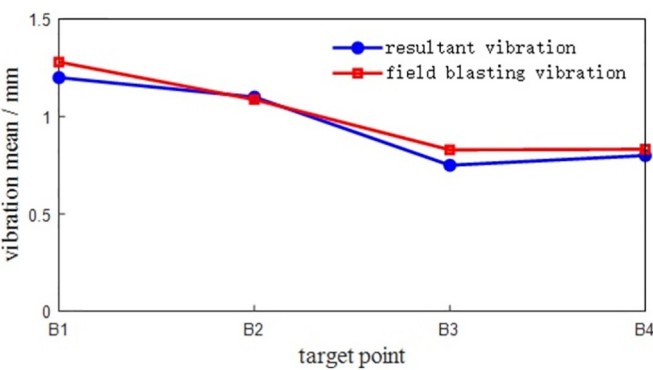

**Fig 13.**

When the external excitation frequency is close to its natural frequency, the viscous damping of the turntable and the fuselage will play a role in suppressing resonance.

## Analysis of OTPA results

The vibration transmissibility and effective vibration contribution of each excitation point to the turntable are shown in Figs 14 and 15, and the vibration contribution of each excitation point to the hinge point B1 on the turntable is shown in Fig 16.

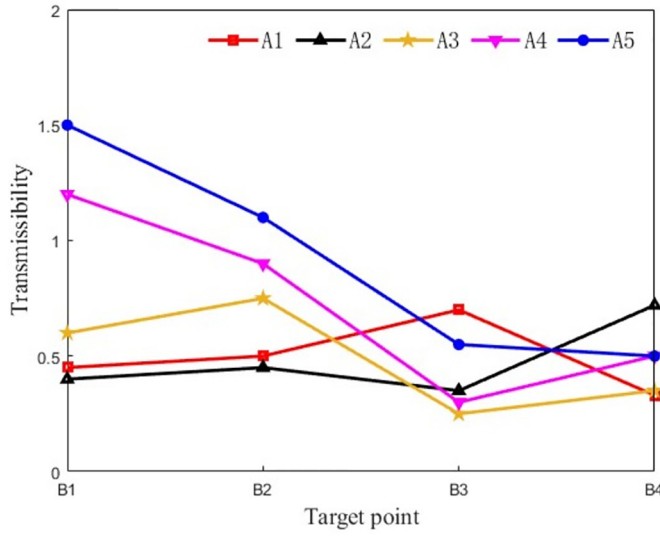

**Fig 14.**

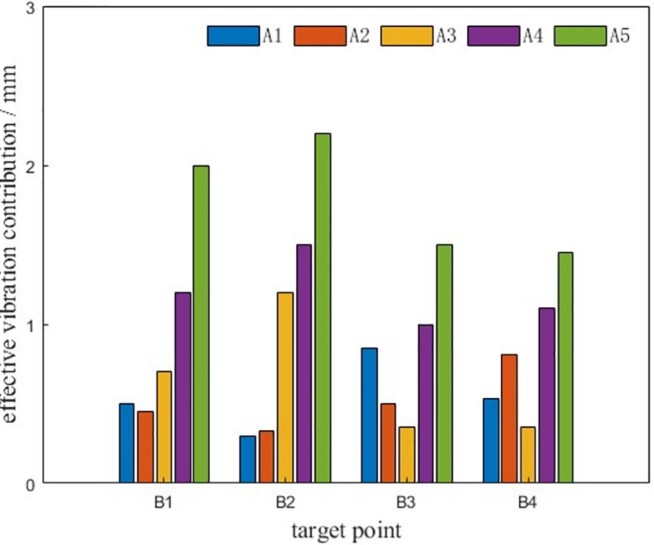

**Fig 15.**

From the diagram, it can be seen that A4 and A5 excitation points have a great influence on the vibration of each hinge ear of the rotary table. The hinge ear B1 on the rotary table is greatly affected by the vibration of A4 and A5 excitation points. The hinge ear B2 under the rotary table is greatly affected by the vibration of A4 and A5 excitation points. The left hinge ear B3 of the rotary table is more sensitive to the vibration of A1 and A5 excitation points. The right hinge ear B4 is more sensitive to the vibration of A2 and A5 excitation points. Among them, the transmissibility of A4 and A5 excitation points to B1 is greater than 1, indicating that there is a vibration amplification effect on the transmission path. The vibration of excitation point A5 has a large contribution to the effective vibration of each trunnion of the rotary table: excitation point A3 only has a large contribution to the effective vibration of B2,

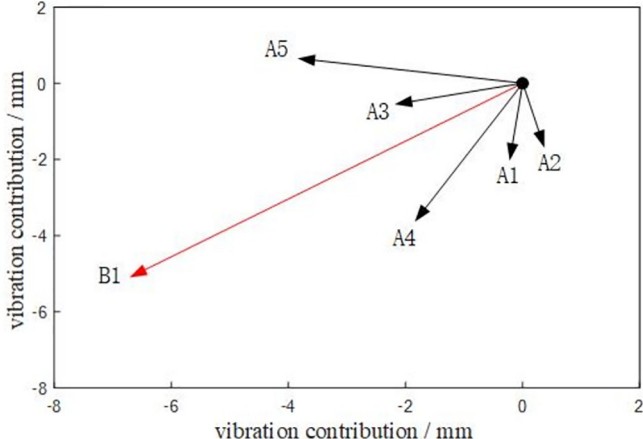

**Fig 16.**

excitation point A1 only has a large contribution to the effective vibration of B3, excitation point A2 only has a large contribution to the effective vibration of B4, and A1-A3 has a small contribution to the vibration of each trunnion of the rotary table.

## Rotary table damage and life prediction based on finite element method

### The basic process of rotary table life prediction

The basic process of finite element analysis of random vibration fatigue life of rotary table is shown in Fig 17. Firstly, an effective finite element model of the rotary table structure is established, and the modal shape of the single rotary table mechanism is simulated to determine the stress concentration position of the rotary table during the cutting process. Based on the frequency response analysis, the frequency response function (FRF) of the stress concentration position of the rotary table is obtained, and the power spectral density (PSD) of the stress response at the dangerous position of the rotary table is extracted. Combined with the S-N curve of the rotary table, the random vibration fatigue life of the rotary table is obtained based on the frequency domain analysis method.

The structure and size of the rotary table model for random vibration fatigue finite element analysis are shown in Fig 18. The model material is ordinary Q235 steel, and the material parameters are shown in Table 2. The S-N curve of the turntable is shown in Fig 19.

In order to maintain consistency with the vibration test experiment of the rotary table, the three-dimensional model of the rotary table was established by using SolidWorks three-dimensional modeling software 1: 1. Based on the Modal modal analysis module of ANSYS workbench simulation software, the material properties (Materials) of the parts were set as structural steel. The 20-node Solid186 hexahedron grid (Hex Dominant) was used to divide,

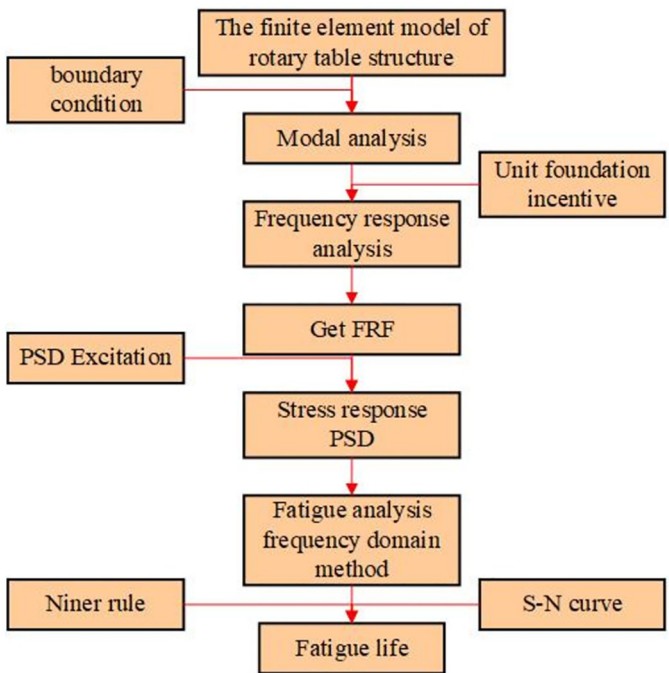

**Fig 17.**

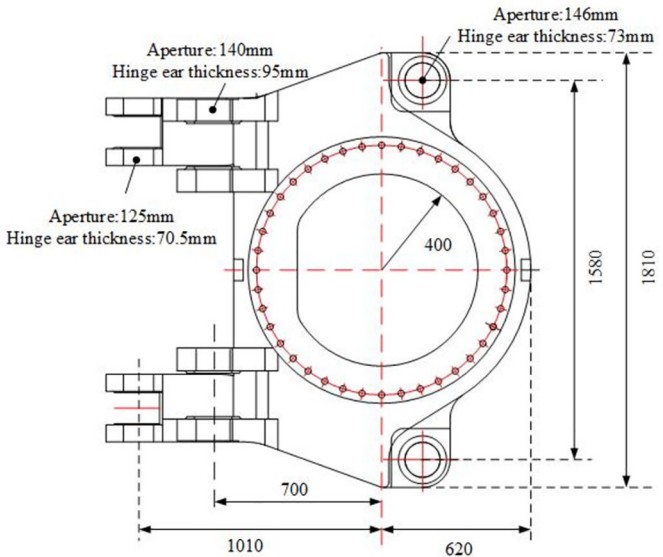

**Fig 18.**

**Table 2. Material parameters of Q235.**

| material | elastic modulus /Gpa | density /(kg/s⁻³) | poisson ratio | ultimate strength /MPa |
|---|---|---|---|---|
| Q235 | 200.0 | 7850.0 | 0.3 | 440.0 |

and the rotary center of the rotary table was used as the fixed support. The number of grids was 5880. The meshing results and boundary conditions are shown in Fig 20, and the modal shapes of the rotary table are shown in Fig 21.

The modal method is used to analyze the frequency response of the model in the frequency range of 10 ~ 500Hz. The unit acceleration load is applied to the rotary table structure, and the

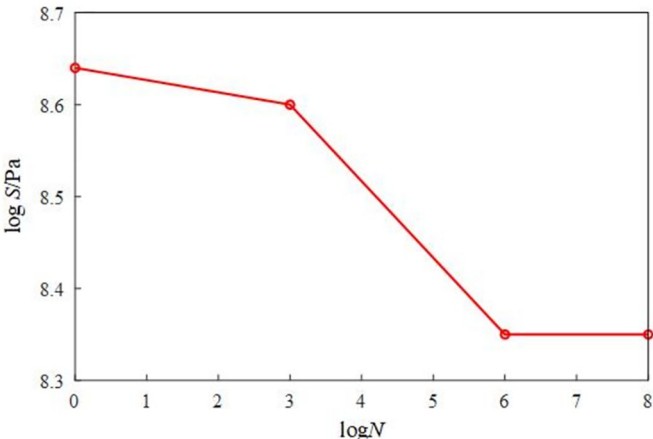

**Fig 19.**

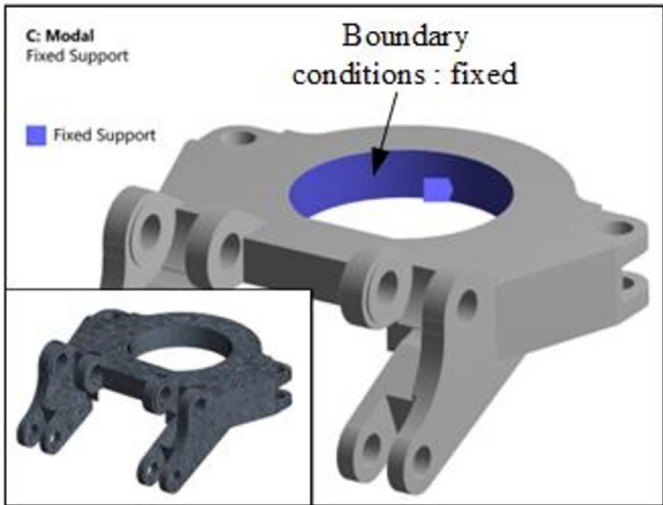

**Fig 20.**

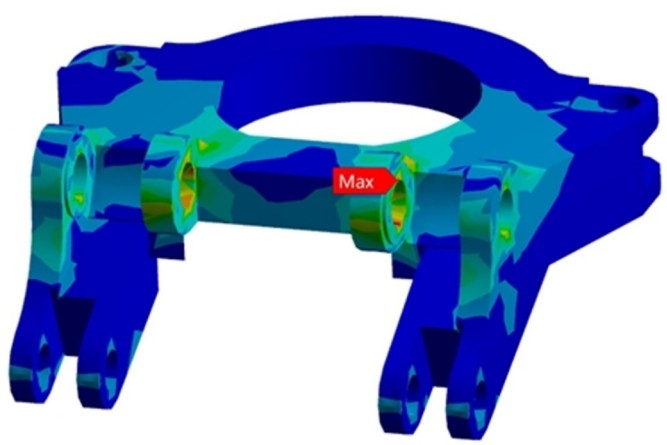

**Fig 21.**

acceleration excitation spectrum is input. The excitation frequency range is set to 10 ~ 500Hz, the excitation amplitude is 0.02g²/Hz, and the excitation root mean square value is 3.135g. The PSD curve at the maximum stress of the rotary table is extracted as shown in Fig 22, and the excitation root mean square value is 75.64MPa.

The above simulation results and the load spectrum file of the turntable are input into ANSYS nCode software to solve the fatigue life of the turntable under symmetrical cyclic load, as shown in Fig 23.

From the diagram, it can be seen that the hinge ear on the rotary table is subjected to a large force, which is most prone to fatigue failure. The minimum number of load cycles is 38160 times. It is assumed that the roadheader runs 20 hours a day, and the time for cutting the section once per cycle is 2 hours. Under ideal conditions, the rotary table can run for 10.45years.

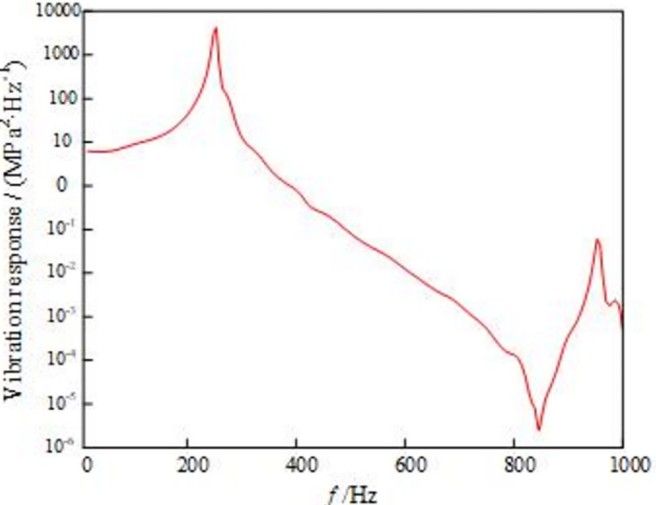

**Fig 22.**

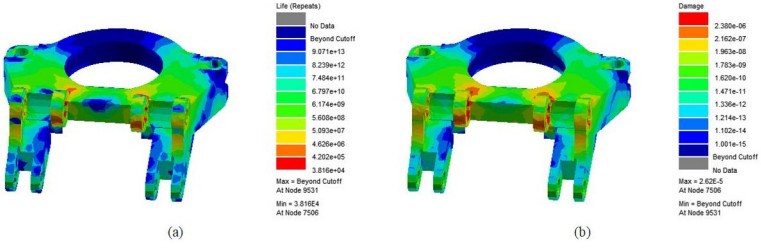

**Fig 23.**

## Rotary table multi-objective optimization

From the above analysis, it can be seen that the main force-bearing part of the rotary table under working condition is a hinge ear. Therefore, the structure of the upper hinge ear, the lower hinge ear and the rotary hinge ear of the rotary table is optimized. By optimizing the specific size of the hinge ear hole, the maximum equivalent stress and the maximum deformation are reduced, so as to improve the reliability of the rotary table and improve the service life.

Based on the response surface optimization analysis method, the size of the aperture of the hinge ear of the rotary table is optimized. There are many kinds of experimental design methods available in the response surface model construction. Box-Behnken Design (BBD) method is a classical and commonly used experimental design method. The response surface model of this method can select 2 to 5 factors; the basic principle is: based on the sampling point response data, by constructing a polynomial function, and then to fit its implicit function. Considering that the actual response cannot be obtained at the current stage, it is assumed that the output of the finite element simulation is the system response, which has the following relationship with the implicit function f (x):

$$Y = f(x) \tag{13}$$

Assuming that f (x) is obtained by the second-order polynomial regression method, the system response is:

$$Y = \beta_0 + \sum_{i=1}^{k} \beta_i x_i + \sum_{i=1}^{k} \beta_{ii} x_i^2 + \sum_{i,j=1}^{k} \beta_{ij} x_i x_j + \varepsilon \tag{14}$$

In the formula: $Y$ is the optimization objective, $\beta_0$ is the fixed value, $\beta_i$, $\beta_i$, $\beta_{ij}$ are the optimization coefficients, $x_i$, $x_j$ are the optimization variables, $\varepsilon$ is the random error.

The multi-objective optimization process of the turntable is shown in Fig 24. The optimization process is to first determine the optimization target as the rotary table component of the roadheader, determine the optimization parameters of the rotary table, and then calculate the sensitivity of the optimization parameters. According to the BBD, the experimental selection points are selected, and the performance of the rotary table is calculated by the finite element method. The response surface model is constructed, and the parameters of the rotary table are optimized based on the model.

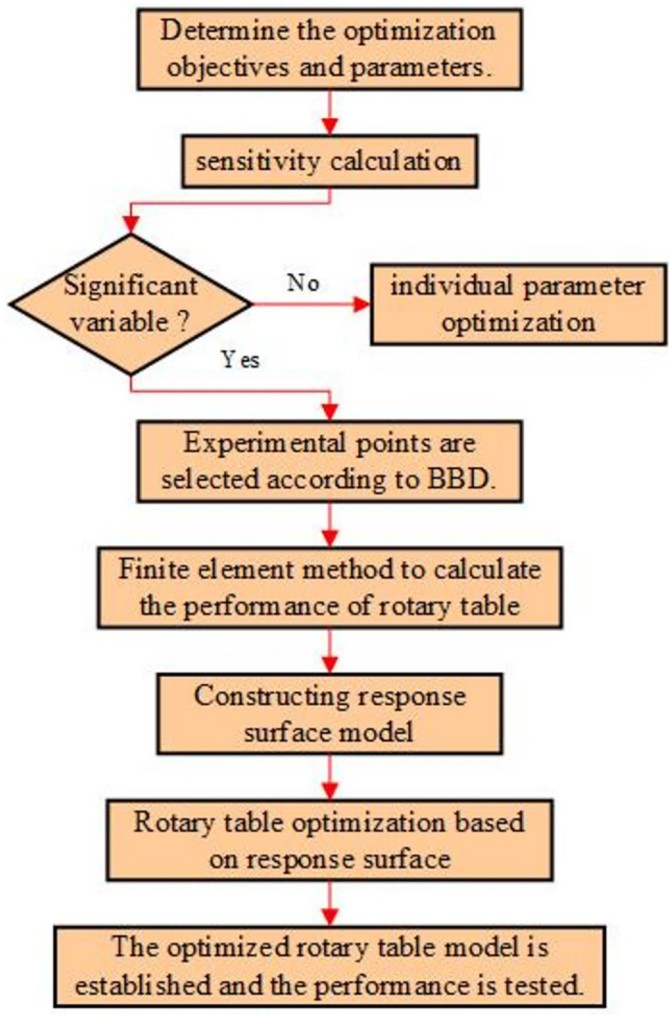

**Fig 24.**

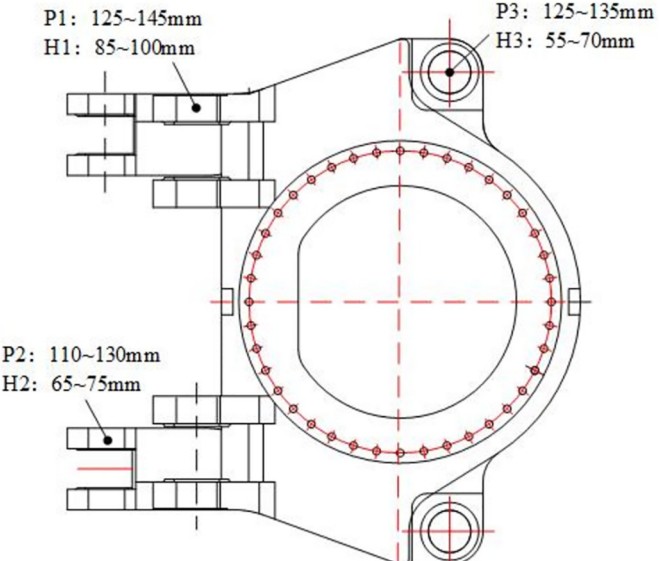

**Fig 25.**

The maximum stress $I_1$ and the maximum strain $I_2$ of the rotary table are set as the optimization objectives. In order to study the influence of optimization parameters on the above optimization objectives, a total of six optimization parameters, including the diameter of the upper hinge ear $P_1$, the diameter of the lower hinge ear $P_2$, the diameter of the rotary hinge ear $P_3$, the thickness of the upper hinge ear $H_1$, and the thickness of the lower hinge ear $H_2$, are selected as the optimization parameters, as shown in Fig 25. The range of optimization parameters is shown in Table 3.

Considering the large number of initial optimization parameters, the number of multi-objective optimization samples is complicated and the influence analysis is carried out. In order to select the optimization space reasonably, the parameter sensitivity analysis is carried out. The sensitivity index of the optimization parameters is:

$$\bar{S}_{ni}^{m} = \frac{\partial f}{\partial z_i}\Big|_{NOP} \frac{z_i}{f} \simeq \frac{\Delta f/f}{\Delta z_i/z_i} \tag{15}$$

In the formula: z is the optimization parameter and f is the optimization objective function.

In order to facilitate observation and comparison, the calculation results are normalized, and the calculation results are shown in Fig 26.

**Table 3. Range of design variables of rotary table.**

| variable name | parameter | minimum value /mm | maximum value /mm |
| --- | --- | --- | --- |
| Upper hinge ear aperture | P1 | 125 | 145 |
| Lower hinge ear aperture | P2 | 110 | 130 |
| Rotary hinge ear aperture | P3 | 125 | 135 |
| Upper hinge ear thickness | H1 | 85 | 100 |
| Thickness of lower hinge ear | H2 | 65 | 75 |
| Rotary hinge ear thickness | H3 | 55 | 70 |

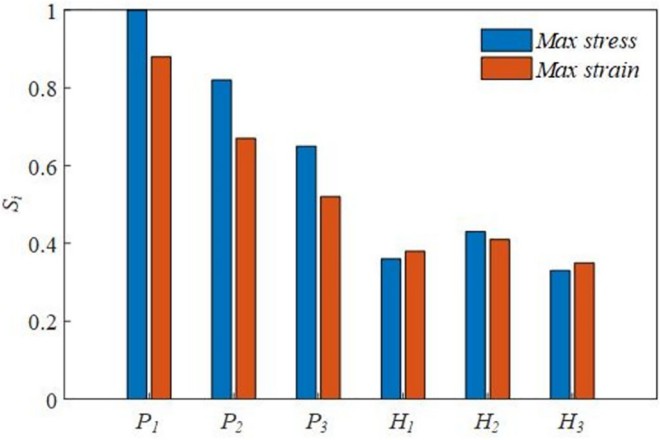

**Fig 26.**

Compared with other optimization parameters, P1, P2 and P3 are more significant and need to be further optimized. H1, H2 and H3 are non-significant values, which can be determined by parameter optimization.

## Analysis of optimization results

Based on the finite element software, the simulation experiment is carried out, and the second-order regression model is established by Design Expert. The aperture of each hinge ear of the rotary table and its maximum stress and maximum deformation response curve are shown in Fig 27.

In order to reduce the maximum equivalent stress and the maximum deformation, based on the response surface optimization method, three sets of optimal alternative solutions are automatically generated. The details are shown in Table 4 below. The comparative analysis of the results before and after size optimization is shown in Table 5:

From the results, it can be seen that the change of the aperture of the rotary hinge ear can affect the maximum stress and maximum deformation of the rotary table. Among them, the change of the aperture of the upper hinge ear has the most obvious influence on its stress and strain. The reason is that the upper hinge ear connects the cutting part as the main force part. The vibration of the cutting head during the cutting process is mainly transmitted from the cutting part to the upper hinge ear of the rotary table. The maximum equivalent stress of the optimized rotary table is reduced by 15.82%, and the maximum deformation is reduced by 24.70%. The optimized structure can better adapt to the cutting process, improve the service life of the rotary table, and achieve the purpose of optimizing the structure of the rotary table.

## Conclusion

A vibration source analysis method of rotary table based on condition transfer path analysis is proposed. The vibration data of cutting head, cantilever, lifting and rotating hydraulic cylinder under stable cutting condition are used as input and output signals. The condition transfer path analysis model of rotary table is established. The vibration source of rotary table in cutting process is analyzed. The contribution and contribution degree of each excitation point of cutting system to rotary table vibration are studied. The main excitation source of rotary table

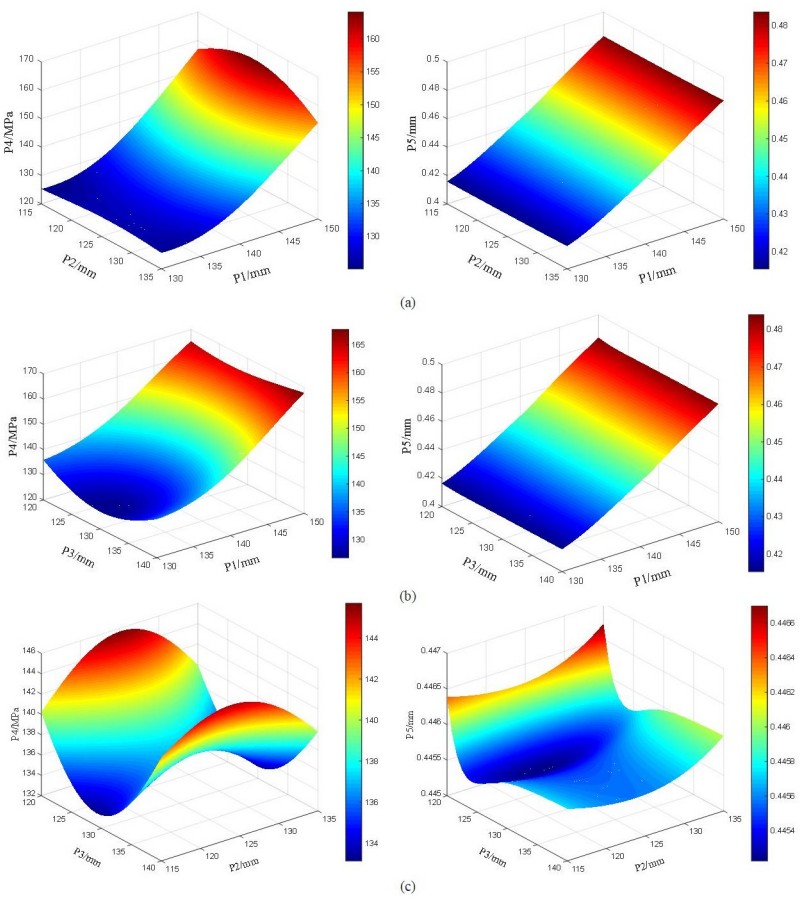

**Fig 27.**

**Table 4. Rotary table optimization results.**

| variable name | Candidate point 1 | Candidate point 2 | Candidate point 3 | Candidate point 4 |
|---|---|---|---|---|
| Upper hinge ear aperture | 128.12 | 130.33 | 133.06 | 140.35 |
| Lower hinge ear aperture | 112.01 | 116.84 | 121.45 | 125.75 |
| Rotary hinge ear aperture | 128.11 | 129.30 | 128.72 | 133.72 |
| Maximum stress | 115.23 | 125.56 | 131.18 | 134.18 |
| Maximum deformation | 0.33564 | 0.41572 | 0.42577 | 0.43577 |

**Table 5. Comparative analysis of results before and after optimization.**

| target variable | primary parameter | optimization parameters | results comparison |
|---|---|---|---|
| maximum stress /MPa | 136.89 | 115.23 | -15.82% |
| maximum deflection /mm | 0.4457 | 0.3356 | -24.70% |

vibration is explored. The vibration test experimental platform is built to test the vibration response of each part of rotary table in cutting process, and the feasibility of this method is verified. The finite element simulation calculation of the rotary table is carried out, and its life and damage cloud map are obtained. Based on the response surface optimization analysis method, the multi-objective optimization design of the key force points of the rotary table is carried out. The maximum equivalent stress of the optimized rotary table is reduced by 15.82%, and the maximum deformation is reduced by 24.70%. The optimized structure can better adapt to the cutting process, improve the service life of the rotary table, and achieve the purpose of optimizing the structure of the rotary table. The research results are beneficial to enrich the research theory of rotary table vibration, and to improve the service life of rotary table and the efficiency of excavation and mining.

## Author Contributions

**Conceptualization:** Miao Xie, He Wang.

**Data curation:** Miao Xie, He Wang.

**Formal analysis:** He Wang.

**Funding acquisition:** He Wang.

**Investigation:** Miao Xie, He Wang.

**Methodology:** He Wang.

**Project administration:** He Wang, Zhixiang Liu.

**Resources:** He Wang.

**Software:** He Wang.

**Supervision:** He Wang, Suning Ma, Xia Wu, Yufeng Dong.

**Validation:** He Wang.

**Visualization:** He Wang.

**Writing – original draft:** He Wang.

**Writing – review & editing:** He Wang.

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
