## [Decision Letter · Decision Letter 0]

23 Apr 2024

PONE-D-24-09048Vibration source analysis and structural optimization design of rotary table based on OTPAPLOS ONE

Dear Dr. HE,

Thank you for submitting your manuscript to PLOS ONE. After careful consideration, we feel that it has merit but does not fully meet PLOS ONE’s publication criteria as it currently stands. Therefore, we invite you to submit a revised version of the manuscript that addresses the points raised during the review process.

We look forward to receiving your revised manuscript.

Kind regards,

Jianguo Wang, PhD

Academic Editor

PLOS ONE

Journal Requirements:

   "This work was supported by the National Natural Science Foundation of China ( 51874158 ), the National Natural Science Foundation of China Youth Science Foundation ( 51904142 ), "

Reviewers' comments:

Reviewer's Responses to Questions

**Comments to the Author**

1. Is the manuscript technically sound, and do the data support the conclusions?

Reviewer #1: Yes

Reviewer #2: Yes

2. Has the statistical analysis been performed appropriately and rigorously? 

Reviewer #1: Yes

Reviewer #2: Yes

3. Have the authors made all data underlying the findings in their manuscript fully available?

Reviewer #1: Yes

Reviewer #2: No

4. Is the manuscript presented in an intelligible fashion and written in standard English?

Reviewer #1: Yes

Reviewer #2: Yes

5. Review Comments to the Author

Reviewer #1: The study addresses an important problem, and it contains an interesting and in-depth mathematical, experimental and simulation analysis regarding stability during cutting process of the boom-type roadheader. The authors have collected a unique dataset using cutting edge methodology. Hence, the proposed method in this paper can be useful precursors of further progress in this field. Overall, the manuscript is well thought out and written, the objectives clearly stated, applied methods are advanced, data statistically analyzed, the conclusions well supported by the data presented. The reviewer recommends its acceptance after considering some relatively minor corrections/clarifications improving the clarity of presentation as follows:

1. The article is based on the vibration data under steady cutting condition as input and output signals, and further explanation on the selection of working conditions is suggested (smooth working conditions are used in the theoretical analysis part, and extreme working conditions are used in the fatigue analysis part). The author needs to unify the working mode of manuscript research.

2. The specifications of the accelerometer are missing. Please check.

3. The manuscript carried out response surface optimization for the rotary table ; considering that the number of initial optimization parameters is large and the number of multi-objective optimization samples is complicated, it is suggested that the author carry out parameter sensitivity analysis before optimization.

4. Manuscript references need to be further standardized ; it is suggested that the author unify the reference format of the manuscript to meet the requirements of the journal format.

Reviewer #2: By establishing a spatial mechanical model of the cantilever roadheader, the variation law of the load at the hinge ear of the rotary table is obtained; Based on the analysis method of working condition path transfer, establish a model for analyzing the working condition transfer path of the rotary table, simulate the vibration of the rotary table during the cutting process, and determine the main vibration source of the rotary table; Furthermore, the correctness of the theoretical model was verified through cutting experiments; Finally, finite element analysis combined with response surface optimization was used to optimize its structure, improve the service life of the rotary table and enhance its operational stability.

1.under ideal conditions, the rotary table can run for 10.45 years. Not 10.6!

2.it can be seen from the curve results that the maximum positive amplitude of the turntable is not 3.4751mm!

3.the hinge ear B2 under the rotary table is greatly affected by the vibration of A4 and A5 excitation points. Not A3 and A5!

In addition, it is not that A3 and A5 lack explanations for maximum stress I1 and maximum strain I2.

4.in the finite element analysis section, the location of adding boundary conditions is not shown in the image.

5.only solve the fatigue life of the turntable under symmetric cyclic loading; Lack of research under irregular load conditions.

6.for the sake of consistency, replace "turntable" with "rotary table" in the title of Table 1.

7. Missing keyword section, formula not centered, paragraph not indented on first line, and formula numbers is chaotic.

6. PLOS authors have the option to publish the peer review history of their article (what does this mean?). If published, this will include your full peer review and any attached files.

Reviewer #1: No

Reviewer #2: No

---

## [Author Response · Author response to Decision Letter 0]

25 Apr 2024

Reviewer #1: The study addresses an important problem, and it contains an interesting and in-depth mathematical, experimental and simulation analysis regarding stability during cutting process of the boom-type roadheader. The authors have collected a unique dataset using cutting edge methodology. Hence, the proposed method in this paper can be useful precursors of further progress in this field. Overall, the manuscript is well thought out and written, the objectives clearly stated, applied methods are advanced, data statistically analyzed, the conclusions well supported by the data presented. The reviewer recommends its acceptance after considering some relatively minor corrections/clarifications improving the clarity of presentation as follows:

1. The article is based on the vibration data under steady cutting condition as input and output signals, and further explanation on the selection of working conditions is suggested (smooth working conditions are used in the theoretical analysis part, and extreme working conditions are used in the fatigue analysis part). The author needs to unify the working mode of manuscript research.

Reply : It has been modified and marked red in the text, please check ! The analysis of the previous and subsequent chapters of this paper is carried out under the same working condition, that is, the theoretical analysis and experimental analysis under the steady-state cutting condition.

2. The specifications of the accelerometer are missing. Please check.

Reply : It has been modified and marked red in the text, please check !

3. The manuscript carried out response surface optimization for the rotary table ; considering that the number of initial optimization parameters is large and the number of multi-objective optimization samples is complicated, it is suggested that the author carry out parameter sensitivity analysis before optimization.

Reply : This part has been added and marked red in the text, please check !

4. Manuscript references need to be further standardized ; it is suggested that the author unify the reference format of the manuscript to meet the requirements of the journal format.

Reply : It has been modified and marked red in the text, please check !

 

Reviewer #2: By establishing a spatial mechanical model of the cantilever roadheader, the variation law of the load at the hinge ear of the rotary table is obtained; Based on the analysis method of working condition path transfer, establish a model for analyzing the working condition transfer path of the rotary table, simulate the vibration of the rotary table during the cutting process, and determine the main vibration source of the rotary table; Furthermore, the correctness of the theoretical model was verified through cutting experiments; Finally, finite element analysis combined with response surface optimization was used to optimize its structure, improve the service life of the rotary table and enhance its operational stability.

1.under ideal conditions, the rotary table can run for 10.45 years. Not 10.6!

Reply : It has been modified and marked red in the text, please check !

2.it can be seen from the curve results that the maximum positive amplitude of the turntable is not 3.4751mm!

Reply : It has been modified and marked red in the text, please check ! The maximum positive amplitude of the turntable is 3.5083 mm.

3.the hinge ear B2 under the rotary table is greatly affected by the vibration of A4 and A5 excitation points. Not A3 and A5!In addition, it is not that A3 and A5 lack explanations for maximum stress I1 and maximum strain I2. 

Reply : It has been modified and marked red in the text, please check !

4.in the finite element analysis section, the location of adding boundary conditions is not shown in the image. 

Reply : It has been modified and marked red in the text, please check ! The rotation center of the rotary table is taken as the fixed constraint.

5.only solve the fatigue life of the turntable under symmetric cyclic loading; Lack of research under irregular load conditions. 

Reply : The main goal of this paper is to optimize the structure of the rotary table and reduce the maximum stress and maximum deformation during the working process of the rotary table. One of the loading conditions ( symmetrical cyclic loading ) is selected to study the fatigue life of the rotary table. The problem of vibration fatigue of the rotary table under irregular load conditions can be considered in subsequent research.6.for the sake of consistency, replace "turntable" with "rotary table" in the title of Table 1. 

6.for the sake of consistency, replace "turntable" with "rotary table" in the title of Table 1.

Reply : It has been modified and marked red in the text, please check !

7. Missing keyword section, formula not centered, paragraph not indented on first line, and formula numbers is chaotic. 

Reply : It has been modified and marked red in the text, please check !

---

## [Decision Letter · Decision Letter 1]

30 May 2024

Vibration source analysis and structural optimization design of rotary table based on OTPA

PONE-D-24-09048R1

Dear Dr. HE,

We’re pleased to inform you that your manuscript has been judged scientifically suitable for publication and will be formally accepted for publication once it meets all outstanding technical requirements.

Kind regards,

Jianguo Wang, PhD

Academic Editor

PLOS ONE

Additional Editor Comments (optional):

Reviewers' comments:

Reviewer's Responses to Questions

**Comments to the Author**

1. If the authors have adequately addressed your comments raised in a previous round of review and you feel that this manuscript is now acceptable for publication, you may indicate that here to bypass the “Comments to the Author” section, enter your conflict of interest statement in the “Confidential to Editor” section, and submit your "Accept" recommendation.

Reviewer #1: (No Response)

Reviewer #2: All comments have been addressed

2. Is the manuscript technically sound, and do the data support the conclusions?

Reviewer #1: (No Response)

Reviewer #2: Yes

3. Has the statistical analysis been performed appropriately and rigorously? 

Reviewer #1: (No Response)

Reviewer #2: Yes

4. Have the authors made all data underlying the findings in their manuscript fully available?

Reviewer #1: (No Response)

Reviewer #2: Yes

5. Is the manuscript presented in an intelligible fashion and written in standard English?

Reviewer #1: (No Response)

Reviewer #2: Yes

6. Review Comments to the Author

Reviewer #1: (No Response)

Reviewer #2: (No Response)

7. PLOS authors have the option to publish the peer review history of their article (what does this mean?). If published, this will include your full peer review and any attached files.

Reviewer #1: No

Reviewer #2: No

---

## [Editor Report · Acceptance letter]

4 Jun 2024

PONE-D-24-09048R1 

PLOS ONE

Dear Dr. WANG, 

I'm pleased to inform you that your manuscript has been deemed suitable for publication in PLOS ONE. Congratulations! Your manuscript is now being handed over to our production team.

Kind regards, 

on behalf of

Dr. Jianguo Wang 

Academic Editor

PLOS ONE